# Identification and Functional Analysis of Tomato *TPR* Gene Family

**DOI:** 10.3390/ijms22020758

**Published:** 2021-01-13

**Authors:** Xi’nan Zhou, Yangyang Zheng, Zhibo Cai, Xingyuan Wang, Yang Liu, Anzhou Yu, Xiuling Chen, Jiayin Liu, Yao Zhang, Aoxue Wang

**Affiliations:** 1College of Life Sciences, Northeast Agricultural University, Harbin 150030, China; Xinan15245116526@163.com (X.Z.); Caizhibo2019@outlook.com (Z.C.); wxy18246718896@163.com (X.W.); 18245699439@163.com (A.Y.); 2College of Plant Protection, China Agricultural University, Beijing 100000, China; hsqjhj@163.com (Y.Z.); yangliu_neau@foxmail.com (Y.L.); 3College of Horticulture and Landscape Architecture, Northeast Agricultural University, Harbin 150030, China; chenx@neau.edu.cn; 4College of Sciences, Northeast Agricultural University, Harbin 150030, China; 13040216@163.com

**Keywords:** bioinformatics, disease resistance, molecular mechanism, tomato, *TPR* gene family, *SlTPR4*

## Abstract

Tomato (*Solanum lycopersicum*) as an important vegetable grown around the world is threatened by many diseases, which seriously affects its yield. Therefore, studying the interaction between tomato and pathogenic bacteria is biologically and economically important. The *TPR* (Tetratricopeptide repeat) gene family is a class of genes containing *TPR* conserved motifs, which are widely involved in cell cycle regulation, gene expression, protein degradation and other biological processes. The functions of *TPR* gene in *Arabidopsis* and wheat plants have been well studied, but the research on *TPR* genes in tomato is not well studied. In this study, 26 *TPR* gene families were identified using bioinformatics based on tomato genome data, and they were analyzed for subcellular localization, phylogenetic evolution, conserved motifs, tissue expression, and GO (Gene Ontology) analysis. The qRT-PCR was used to detect the expression levels of each member of the tomato *TPR* gene family (*SlTPRs*) under biological stress (*Botrytis cinerea*) and abiotic stress such as drought and abscisic acid (ABA). The results showed that members of the tomato *TPR* family responded to various abiotic stresses and *Botrytis cinerea* stress, and the *SlTPR2* and *SlTPR4* genes changed significantly under different stresses. Using VIGS (Virus-induced gene silencing) technology to silence these two genes, the silenced plants showed reduced disease resistance. It was also shown that *TPR4* can interact with atpA which encodes a chloroplast ATP synthase CF1 α subunit. The above results provide a theoretical basis for further exploring the molecular mechanism of *TPR*-mediated resistance in disease defense, and also provide a foundation for tomato disease resistance breeding.

## 1. Introduction

Tomato is an important vegetable utilized globally for its nutritional and medicinal values. At the same time, tomato is also an important model plant for studying plant evolution, genetics, and resistance to stress. Tomato has the ability to resist adverse external environment due to a long-term evolutionary history, but it is susceptible to various pathogenic bacteria which hampers its growth and development [1,2,3].

In plants, many NB-LRR (Nucleotide Binding, Leucine Rich Repeat) proteins which contains a C-terminal LRR domain, a variable N-terminal effector domain, and a central NB domain belong to *TPR* (Tetratricopeptide repeat) gene family which is a class of genes that contain *TPR* conserved motifs and is widely distributed in nature. *TPR* often exists as a right-handed supercoil structure and has an amphipathic channel. The existence of this channel enables *TPR* to adapt and complement the target protein interaction region [4]. *TPR* consists of multiple *TPR* conserved motifs containing 34 amino acids, each *TPR* conserved motif consists of two antiparallel α-helix subunits (helixA, helixB). Since the first discovery of *TPR* repeats in yeast, more than 20,000 functionally unrelated *TPR* proteins have been identified [5,6]. This gene family is widely involved in biological processes such as cell cycle regulation, gene expression, protein transport, and protein degradation [5]. In microorganisms and animals, *TPR* mutations are associated with the virulence of *Borrelia burgdorferi* and pathogenesis of diseases [7,8,9,10,11]. In addition, the TPR conserved motif can inhibit the release of virus particles and regulate mitochondrial division [12,13]. *TPR* repeats are also found in many plants and can mediate protein-chaperone interactions. Studies have found that many *TPR*s are involved in plant stress and hormone signaling. It was found that NCA1 interacts with CATALASE2 (CAT2) through its C-terminal *TPR* repeat sequence, which is necessary for maintaining H_2_O_2_ homeostasis in plant cells. In the presence of NCA1, catalase activity increases sharply. At this time, zinc ions bind to its N-terminal zinc finger domain to maintain its functional state in response to the oxidation process [14]. SPINDLY (SPY) containing *TPR* repeats was used as a negative regulator of GA (Gibberellins) signals, and its deletion mutants showed insensitivity to cytokinin, indicating that SPY plays a role in two hormone signaling pathways. And it is speculated that SPY mediates interactions with other regulatory proteins through its *TPR* repeats and participates in other regulatory pathways [15]. In addition to plant hormone responses, *TPR* also plays a role in stress regulation. A new plant-specific TPR protein, TTL1 (TETRATRICOPEPTIDE-REPEAT THIOREDOXIN-LIKE 1) can positively regulate the stress response regulated by ABA. The loss of TTL1 function causes plants to be sensitive to salt and osmotic stress during seed germination and later development [16]. Kwon et al. found that SRFR1, a *TPR* repeat protein, mediate plant immune response triggered by the effector molecule AvrRps4 [15]. Deletion of SRFR1 makes plants susceptible to diseases [17]. In addition, the chaperone heat shock proteins Hsp70 (Heat Shock Protein 70) and Hsp90 (Heat Shock Protein 90) interact with many chaperones containing *TPR* motifs to participate in ETI (Effector-Triggered Immunity) response. The molecular chaperone protein STI1 (Suppressor of the G2 Allele of skp1) is a stress protein. It was first discovered in yeast that STI1 is a component of the multi-protein chaperone complex of Hsp90 and Hsp70, which can function as a “bridge” between two chaperones [18,19,20]. The TPR repeat of *Arabidopsis* TPR protein FLU (Fluorescent) can interact with aminoacyl-tRNA reductase, thereby regulating the production of δ-aminopropionyl acetate, a key substance in the process of chlorophyll synthesis [21].The SPINDLY protein containing TPR repeats in *Arabidopsis* can interact with GIGANTEA protein to regulate biological processes such as flowering, biological rhythm, and hypocotyl elongation [22]. Schweiger et al. found that *Arabidopsis* TPR7 contains multiple TPR repeat motifs, and this gene, as an endoplasmic reticulum resident protein, can participate in the nuclear process of post-translationally modified proteins [23].

SGT1 as a member of the TPR family, is highly conserved in eukaryotes. SGT1 and HSP90, RAR1 (Required for MLA12 Resistance 1) can form a complex. On the one hand, it acts as a molecular chaperone and interacts with R protein to mediate the expression of downstream disease resistance genes and changes in related hormone levels. On the other hand, it can maintain the correct folding of R protein and maintain the stability of R protein [24,25]. SGT1 has two members, SGT1a and SGT1b. Studies have reported that SGT1b interacts with Prf protein in tomato through its chaperone activity to stabilize the Prf protein, thereby contributing to Prf-mediated resistance to *Pseudomonas syringae* [26]. Silencing SGT1 in tobacco can reduce the stability of the R protein and resistance to pathogens [27].

ATP synthase is a key enzyme in energy metabolism. It mainly exists in the plasma membrane of chloroplasts, mitochondria and bacteria. It can synthesize ATP using the proton gradient produced by oxidative phosphorylation (mitochondria) and photosynthesis (chloroplast) [28]. The *atpA* gene encodes the CF1 α subunit of chloroplast ATP synthase. atpA not only plays an important role in regulating the interaction between enzymes and reaction substrates, but also in the interaction between biological organisms and the external environment. In tomato, ATP synthase can respond to the fusiformin secreted by pathogens, thereby causing the accumulation of intracellular salicylic acid and the expression of disease resistance-related genes [29]. Recent studies have found that the effector RipX can interact with mitochondrial atpA, and induce plant defense response by inhibiting atpA expression [30]. In conclusion, ATP synthase and TPR family have been reported to participate in disease resistance and defense response, but the relationship between them has rarely been reported.

Although *TPR* genes have been studied in *Arabidopsis*, wheat and other plants, a large number of them remain uncharacterized. The function of *TPR* gene family members in tomato remain elusive [31,32,33,34,35]. This study is based on the tomato genome, using bioinformatics to analyze the number of tomato *TPR* gene family members, gene structure, system evolution, participation pathways, expression patterns, protein interactions, etc. qRT-PCR was used to examine the changes in the expression of each member under various stress conditions. Virus-induced over-gene silencing technology was used to further verify the function of tomato *SlTPR4* and *SlTPR2* genes. Because changes in *TPR4* gene expression is most significant under biotic stress, we focused on *TPR4*. We explore the mechanism of defense response it mediates and found that it can interact with atpA. The above research results will provide a theoretical basis for further exploring the molecular mechanism of *TPR*-mediated disease resistance and defense response. These provide new ideas for the in-depth elucidation of the plant-pathogen interaction mechanism and has immense scientific and practical significance in agricultural production.

## 2. Results

### 2.1. Identification and Phylogenetic Analysis of Tomato TPR Genes

The analysis of tomato genome using bioinformatics has identified 26 candidate *TPR* genes (Table 1). It has been observed that the 26 *SlTPR* genes are distributed across all chromosomes except for chromosome X and XII. We named them from chromosome 1 according to their chromosome location. Various physical and chemical properties of the corresponding proteins including their isoelectric point, molecular weight and amino acid number were analyzed using ExPaASy tool. The isoelectric point was found to range from 4.40 (SlTRP1) to 9.5 (SlTRP8). The molecular weight of the proteins was found to range between 13,365 KDa (SlTPR3) and 156,043 KDa (SlTPR6). We use CELLO v2.5 to predict the signal peptides of the amino acid sequences of the members of the *TPR* gene family and presumably the subcellular location. The chromosomal location analysis showed that most genes were localized in the cytoplasm, and a few genes were distributed in the outer membrane, periplasm, and extracellular matrix.

In order to further understand the evolutionary relationship of tomato *TPR* genes, *Arabidopsis thaliana TPRs* was combined with tomato candidate *TPR*s to construct a phylogenetic tree. Between tomato and *Arabidopsis TPR* gene family, 9 pairs of vertical and 7 pairs of parallel homologous gene were found (Appendix A).

### 2.2. Analysis of Tomato TPR Gene Conserved Motif and Gene Structure

The phylogenetic relationship and classification of tomato *TPRs* were supported by motif analysis. We analyzed and predicted the tomato *TPR* gene through MEME software, and predicted 10 conserved motifs (Figure 1). Proteins with similar genetic relationships had similar conserved motifs, indicating that their structures were highly conserved. For example, *SlTPR6* and *SlTPR18*; *SlTPR5* and *SlTPR14* and *SlTPR17* and *SlTPR20* and *SlTPR24* and *SlTPR25*; *SlTPR3* and *SlTPR22*; *SlTPR7* and *SlTPR8*; *SlTPR11* and *SlTPR15* and *SlTPR16*; *SlTPR9* and *SlTPR21*; *SlTPR10* and *SlTPR13* and *SlTPR26*; *SlTPR1* and *SlTPR19* and *SlTPR23*; *SlTPR2* and *SlTPR4* have the same conservative motif and order.

The structural analysis of tomato *TPR* gene by GSDS tool found that most of the family members were intron-enriched genes. Except for the gene *SlTPR19*, there were multiple introns in the remaining genes (Appendix A).

### 2.3. Expression Analysis and GO Analysis of Tomato TPRs Gene

For a more systematic study of the tomato *TPR* gene, expression and GO analysis was performed to predict the role of this gene family in plant growth and development. The study found that the gene family is mainly distributed in the endoplasmic reticulum membrane and cytosol, and a small amount in the nucleus and plasma membrane, and less in the plasmodesma, nuclear membrane and cytoplasm. In terms of molecular function, about 50% of *TPR* proteins are predicted to bind to Hsp90 protein, while other proteins have phosphatase activity. In terms of biological process, they are mainly involved in response to cold and cadmium stress, chloroplast nuclear signal transmission and metabolism of cofactor. In addition, *TPR* is also a component of chaperone-mediated protein complex (Figure 2).

### 2.4. Tissue Expression Analysis of Tomato TPRs Gene

The Genevestigator tomato gene chip platform was used to analyze the expression changes of tomato *TPR* gene family (Figure 3). The results showed that the expression level of genes *SlTPR2* (*solyc01g097190.2*), *SlTPR4*(*Solyc03g007670.2*), *SlTPR10* (*Solyc07g055860.2), SlTPR12* (*Solyc08g079170.2*), *SlTPR14* (*Solyc09g082540.2*), *SlTPR20* (*Solyc03g118690.2)* in the gene chip changes significantly under different conditions. *SlTPR2* (*Solyc01g097190.2*) was expressed in vegetative organs, with the highest expression in pulp and the lowest expression in leaves. In the developmental stage, the gene was mostly expressed in mature fruits and lowest in the flower opening stage. *SlTPR4* (*Solyc03g007670.2*) was also expressed in vegetative organs and developmental stages, but the highest expression was in the lateral roots and root tips. *SlTPR10* (*Solyc07g055860.2*) has a low overall expression level during the development of different vegetative organs, and is not expressed in stems, hypocotyls, flowers, cotyledons, and blade (lamina). The expression of *SlTPR14* (*Solyc09g082540.2*) and *SlTPR20 (Solyc03g118690.2*) are similar, and both are highly expressed in the pericarp walls.

At the same time, we analyzed the biotic stresses of TPR gene family members including rust fungus, and *Botrytis cinerea* treatments, as well as abiotic stress, ABA treatment, traumatic stress, drought, strong light (450–1000 μmol), weak light (450–200 μmol), high temperature, and salt stress changes in expression. Among them, under the stress of rust, the expression of *SlTPR20* was significantly increased. In the *Botrytis cinerea* treatment, *SlTPR2*, *SlTPR4* and *SlTPR14* gene were significantly up-regulated. In ABA treatment, *SlTPR2* and *SlTPR12* gene were significantly up-regulated. In the treatment of wound stress, the expression of *SlTPR2, SlTPR4* and *SlTPR14* was significantly up-regulated. Under drought stress, *SlTPR2* and *SlTPR14* increased significantly. Under strong light and weak light, *SlTPR2, SlTPR4, SlTPR12* and *SlTPR14* increased significantly. In high temperature treatment, *SlTPR2* and *SlTPR12* gene were significantly up-regulated. Under salt stress, the expression of each gene was not significantly up-regulated (Figure 3).

### 2.5. Expression Analysis of Tomato TPR Gene under Stress

Bioinformatics analysis showed that *TPR* family genes respond to various biotic and abiotic stresses. In order to further accurately analyze the function of *TPR*, we treated tomato seedlings with *Botrytis cinerea*, drought and ABA conditions. Gene expression for each gene was detected by qRT-PCR.

Figure 4 shows that expression of *SlTPR2* was significantly up-regulated in *Botrytis cinerea* treatment, and the relative expression level was the highest at 12 h. We define the expression of 0 h as 1, and compare the expression at other time points. It was found that the expression level of the *SlTPR2* at 12 h was about 4.5-fold of that at 0 h. In addition, the expression of *SlTPR4* was also up-regulated at 12 h. Under drought stress, the expression of *SlTPR2* was significantly up-regulated at 3 h, 12 h and 24 h. The expression of *SlTPR12* was only obvious at 24 h, about 9-fold. *SlTPR14* was down-regulated slightly at 0.5 h, but up-regulated at other time points. Under ABA stress, the up-regulated expression of *SlTPR2* and *SlTPR4* was more obvious.

### 2.6. Functional Analysis of Tomato SlTPR2 and SlTPR4 Silencing in Tomato Stress

After prediction by Genevestigator tomato gene chip platform and qRT-PCR results, it was found that under different stresses, the variation in gene expression is most pronounced for *SlTPR2* and *SlTPR4*. *SlTPR2* and *SlTPR4* may play a role in tomato response to stress. To elucidate the biological function of *TPR2* and *TPR4,* we examined its organ-specific expression in tomato by qRT-PCR. *TPR2* was strongly expressed in roots, stems and siliques, but its expression was lower in leaf and flowers. *TPR4* was strongly expressed in leaves (Appendix A). In order to further investigate the role of *SlTPR2* and *SlTPR4* genes accurately, we will take *SlTPR2* and *SlTPR4* as research objects and further verify the gene function through VIGS.

First, we successfully constructed the VIGS vector pTRV2-*SlTPR2*, pTRV2-*SlTPR4*, and transformed them into tomato with pTRV2-PDS, the silencing vector of Phytoene Dehydrogenase (PDS), which was introduced as the indicator gene of VIGS system. The results showed that the silencing efficiency of *SlTPR2* was 36.5% and that of *SlTPR4* was 42.5% (Appendix A). Abiotic and biotic stress can generate excessive ROS production, which are toxic to the cell and result in membrane damage and cell death. Therefore, various physiological indices were measured to evaluate oxidative injury between *SlTPR2*, *SlTPR4* silenced plants, and control plants under *Botrytis cinerea*, drought, and ABA stress.

Under *Botrytis cinerea* stress, we detected the content of SOD (Superoxide dismutase), POD (Peroxidase), PPO (Polyphenol oxidase) and AAO (Ascorbic acid oxidase) in the silenced plants and control plants. Interestingly, the variation trend of SOD content in control plants was opposite to that of *SlTPR2* silenced plants, while that of *SlTPR4* silenced plants was the same. However, the difference is that from 9 h to 12 h, the SOD content of *SlTPR2* silenced plants and control group increased and *SlTPR4* silenced plants decreased. The results showed that the change of AAO content in silence plants and control plants was small, but from 9 h to 12 h, the increase of AAO content was more obvious in *SlTPR4* silenced plants. The variation trend of POD content in control group and *SlTPR2* silenced plants was the same, but it was opposite to that in *SlTPR4* silenced plants from 0 h to 6 h, and the same from 6 h to 12 h. It is worth noting that the variation trend of PPO content in control group was the same as that in *SlTPR4* silenced plants, but opposite to that in *SlTPR2* silenced plants, except from 9 h to 12 h (Figure 5).

Under drought stress, the variation trend of SOD content in the silenced plants was same, but it was a little different from the control group during 0 h to 3 h and 9 h to 12 h. The variation trend of AAO content in the control group and the silenced plants was basically the same, except that *SlTPR2* silenced plants at 9–12 h and *SlTPR4* silenced plants at 3–6 h increased significantly. In addition, the change of POD content was irregular among the control and the two silenced plants, but the change of content of PPO showed the same trend between the control and the two silenced plants (Figure 6).

The content of SOD in the control group and the silencing group fluctuated greatly under ABA stress, and the SOD content in the control group was significantly higher than that in the silence group at 12 h. Similarly, the content of AAO fluctuated greatly. At 6 h, the content of AAO enzyme in the control group was higher than that in the silence group, which indicated that the adaptability of the control group to stress was lower than that in the silence group. The trend of POD content in the control and silent plants was similar. The content of PPO in control group and silence group changed little, but the trend fluctuated obviously (Figure 7).

### 2.7. TPR4 Interacts with atpA

qRT-PCR and gene silencing tests showed that *SlTPR4* plays an important role in disease resistance. In order to further explore the molecular mechanism of *SlTPR4* mediated disease resistance, STRING was used to predict the interaction protein partener of SlTPR4. The protein, atpA, which encodes the ATP synthase α subunit was identified as the interaction partner of SlTPR4. Previous report that ATP/ADP plays an important role in the initiation of ETI responses in plants aroused our interest [36]. In order to accurately identify the interaction between the two proteins, we used yeast two-hybrid to verify the interaction between SlTPR4 and atpA.

*atpA* was cloned into the pGBK vector and *TPR4* was cloned into the pGAD vector. *pGBK-atpA* (BD-atpA) and *pGAD-TPR4* (AD-TPR4) were co transformed into yeast and cultured in SD-Trp-Leu and SD-Trp-His-Leu. If yeast cells grow in SD-Trp-His-Leu medium and turn blue by X-gal, it indicates that there is interaction between the two proteins, otherwise there is no interaction. The yeast two-hybrid assay indicated that positive control group and experimental group could grow and turn blue in SD-Trp-His-Leu medium. In contrast, the pairs of *pGADT7/pGBKT7, pGADT7-TPR4/pGBKT7* and *pGADT7/pGBKT7-atpA* did not have interaction. These results indicated that there was interaction between SlTPR4 and atpA (Figure 8A).

Furthermore, to further verify the interaction between SlTPR4 and atpA protein in vitro, *TPR4*-His and GST-tagged *atpA* were expressed in *E. coli* and purified for pull down assays. First, bind atpA-GST fusion protein or GST protein to Glutathione Sepharose 4B beads, then add SlTPR4-His fusion protein and incubate at 4 °C. The results showed that SlTPR4-His can be combined by atpA-GST, but cannot be pulled down by empty GST protein (Figure 8B). Taken together, these results demonstrate that TPR4 directly interact with atpA in vitro.

## 3. Discussion

This study used bioinformatics to analyze the tomato genome and identified 26 members of the *TPR* family. We use CELLO v2.5 to predict the subcellular localization of *TPR* gene family. It was showed that tomato *TPR* genes are widely distributed on 10 chromosomes. Most of them are localized in the cytoplasm, and a few are localized in the outer membrane and periplasm. Phylogenetic analysis showed that there are 9 pairs of vertically homologous genes in *Arabidopsis* and tomato *TPR* gene families, with high homology. It was showed that the *TPR* genes are relatively conserved during plant evolution [37,38]. GO analysis showed that *TPR* genes, consistent with *Arabidopsis TPR* gene family, are mainly distributed on the cytoplasmic matrix and endoplasmic reticulum membrane. Genevestigator tomato gene chip platform found that the expression of *TPRs* in different developmental stages, tissues and processing conditions were significantly different, which indicates that each *TPR* gene family member has a specific function. Interestingly, *SlTPR2* (*Solyc01g097190.2*) expression was up-regulated under *Botrytis cinerea* treatment, drought stress and ABA stress. Similarly, the expression of *SlTPR4* was also up-regulated under many stress conditions, indicating that these two genes can respond to multiple biotic and abiotic stress conditions. However, how *SlTPR2* and *SlTPR4* respond to a variety of stress conditions and functions need further study.

*TPR* protein may be involved in the transmission of SGT1-induced insect-resistant and disease-resistance pathway signals. The homologous protein of SlTPR4 in *Arabidopsis* is SGT1. In recent years, the function of tomato SGT1 has made progress. COR (coronatine) is produced by several pathogens of *Pseudomonas syringae* and is also an analog of the plant hormone jasmonic acid (JA). Silencing SGT1b in tobacco and tomato leads to reduced cell death and plant chlorosis caused by COR. Studies have suggested that SGT1b is a component of the COR/JA-mediated signal transduction pathway [39,40]. In addition, Bhattarai and other studies have shown that SGT1 can form a complex with the heat shock protein Hsp90 and the NB-LRR disease resistance protein Mi-1 and activate downstream disease resistance pathways, promoting tomato resistance to root-knot nematode and aphid diseases [41]. Studies have found that SGT1 is involved in a variety of NLR protein-mediated anti-disease responses, including Bs2, Bs4, LR2, MLA, Mi, N, Rx, RPS4, Prf, I2 and R3a. Silencing the Hsp90 gene in plants found that Rx, RPM1, Mla, N, Prf, Mi, I2, R3a, Lr21 and RPS2 mediated plant disease resistance weakened indicating that *R* gene-mediated resistance depends on SGT1 and Hsp90 [42].

Previous studies showed that ATPase is widely involved in plant biotic and abiotic stress responses. Yoshida et al. reported that when plants are subjected to chilling stress, their cytoplasmic pH decreases, and ATPase can inhibit cytoplasmic acidification and induce plant cold resistance by regulating the concentration of protons in plant cells [43]. In tomato, ATPase is able to respond to fucoidin secreted by pathogenic bacteria, which causes cell salicylic acid accumulation and disease resistance-related gene expression [29]. Among them, the α subunit of ATPase (*atpA*), is a key subunit of ATPase during ATP synthesis and it is involved in plant resistance to cold, insect and diseases. Koichi et al. showed that overexpression of *atpA* in green beans resulted in an increase resistance to two-spot mite and worm [44]. The above results partially explain the function of *atpA* in the process of plant resistance to cold and disease, but do not explain how *atpA* participates in this process, and its role in the regulatory mechanism.

This study identified *TPR* gene family members through bioinformatics methods. Expression profiling and qRT-PCR show that *SITPR4* gene responds to various stress treatments such as pathogenic bacteria, which indicates that *SITPR4* may play a role as a key gene in the process of abiotic and biotic stress. The study of *SITPR4* gene will help to reveal the molecular mechanism of plant response to biotic and abiotic stress. Studies by Takken et al. have shown that NB-LRR type disease resistance proteins bind and consume ATP during the recognition of an effector protein in order to induce disease resistance [45]. The interaction between tomato Hsp90 and SGT1 mediates the plant resistance to aphids and nematodes [46]. Based on these, we speculate that SITPR4, which is homologous to SGT1 in *Arabidopsis*, also participates in energy metabolism and interacts with atpA. Using pull down and yeast two-hybrid experiments, it was shown that *SITPR4* interacts with atpA, which indicates that SlTPR4 may form a complex with atpA and Hsp90 mediating plant resistance to diseases and insect pests. Therefore, we hypothesize that TPR4 may enter the chloroplast and interact with atpA, thereby binding and depleting ATP in the process of inducing disease resistance.

## 4. Conclusions

Twenty-six *TPR* gene family members in tomato were identified, which were located in the cytoplasm, outer membrane and periplasm. The phylogenetic results showed that *Arabidopsis* and tomato TPR genes have high homology, indicating that the TPR genes are relatively conservative. It was shown that *SlTPRs* are intron-rich gene family by structural analysis. GO analysis predicted that *TPR* genes have molecular chaperone, anti-cold, and Ca^2+^ response functions. Genevestigator gene chip analysis found that the expression of *TPR* gene in various developmental stages and treatments is different.

The *TPR* genes were subjected to various stress including biotic stress (*Botrytis cinerea* stress) and abiotic stress (drought, ABA). The qRT-PCR results were consistent with the predicted results of the gene chip, and it was found that *SlTPR2* and *SlTPR4* strongly responded to multiple stresses. The *SlTPR2* and *SlTPR4* gene silenced plants were subjected to stress treatment and then tested for related enzymes activity. The results showed that the silenced plants were sensitive to biotic and abiotic stress. Yeast two-hybrid and Pull-down experiments found that SlTPR4 interacts with atpA in vitro.

## 5. Materials and Methods

### 5.1. Plant Materials

*S. lycopersicum* “glamor” was supplied by Tomato Genetics Research Center (University of California, Davis, CA, USA) and conserved in our lab. Germination treatment of tomato seeds: Seeds were put in a triangular flask containing 5% sodium hypochlorite and shaked for 15 min, then washed with 75% ethanol once and with sterile water three times, finally they were put in an incubator to germinate. The plant material was sustained in an artificial climate chamber with 16 h light/8 h dark photoperiod and temperature of 25 ±  1 °C. After two days, the germinated seeds were moved to the 9 cm × 9 cm plug tray which contains 3:1 nutrient soil and vermiculite, photoperiod 16 h/8 h (light/dark), temperature 26 °C/21 °C (light/dark). The plant was used for experimental treatment when five leaves appeared on the plant (about a month). The second leaf on the top of the morphology (the second youngest leaf) was taken and stored at −80 °C for freezing. The biological replicate is three times for each treatment.

### 5.2. Bioinformatics Analysis of TPR Gene Family

The TPRs protein sequence was downloaded from the *Arabidopsis* database TAIR, and 36 TPR protein family sequences were performed BLAST homology search in the tomato database SGN (https://solgenomics.net/). We set the E value to 1^e−10^, and removed the duplicates, finally screened the tomato candidate *TPR* family members. The putative TPR protein sequences were submitted to SMART (http://smart.embl-heidelberg.de/) and Pfam (http://pfam.janelia.org/) to confirm the conserved domain. Finally, all of the non-redundant and high-confidence genes were assigned as tomato *TPR*s (*SlTPR*s). These genes were named according to their positions.

Online analysis of the isoelectric point and molecular weight of all TPRs protein amino acids in tomato were used by ExPASy (http://expasy.org). CELLO v2.5 (http://cello.life.netu.edu.tw/) was used to subcellularly locate members of the family and to locate chromosomes through the SGN database.

In order to study the evolutionary relationship of tomato *TPR* genes, MEME (http://meme.nbcr.net/meme/) was used to analyze the conserved domains of *TPR* genes online, and determined up to 10 motifs. ClustalW was used to perform multiple sequence alignments on tomato and *Arabidopsis TPR* gene families, the results were merged into MEGA6.06 software, and the phylogenetic tree of *Arabidopsis* and tomato *TPR* gene families was constructed by Neighbor-Joining. The verification parameter Bootstrap is repeated 1000 times [47].

Genomic and CDS sequence of the obtained tomato *TPR* gene family members were downloaded in the Phytozome (https://phytozome.jgi.doe.gov/pz/portal.html) and analyzed through GSDS (http://gsds.cbi.Pku.edu.cn/).

In order to standardize the analysis of tomato *TPR* gene products, GO analysis of tomato *TPR* gene was performed through AgBase v2.00 (http://agbase.msstate.edu/index.html). Goanna (http://agbase.msstate.edu/GOAnna.html) was used to perform Blastp analysis on tomato protein containing *TPR* gene. Then Goslim Viewer (http://agbase.msstate.edu/GOSlimViewer.html) was used to generate protein function annotation summary. Through the Genevestigator’s tomato gene chip platform (https://genevestigator.com/), the chip data of tomato *TPR* gene family gene expression was analyzed. In the cluster diagram, the depth of red indicated the strength of gene expression.

### 5.3. Quantitative Real Time Polymerase Chain Reaction Analysis

Total RNA was extracted from plants grown at 25 °C for 3 weeks with TRIzol reagent (Invitrogen, USA), and qRT-PCR was performed using the SYBR Green PCR Master Mix kit (TaKaRa, Tokyo, Japan), as previously described [48]. The product cDNA obtained by reverse transcription was diluted 10 times and used as a template. A mixture of 10 uL 2× SYBR Premix ExTaq buffer, 0.3 µL 10 mM 5′/3′-Primer, 0.8 µL DyeII, 2 µL cDNA template, and double distilled water was added to a 96-well PCR plate to a final volume of 20 µL. Then ABI PRISM 7500 real-time quantitative PCR instrument was used for PCR amplification. The reaction condition: 95 °C 30 s; 95 °C 5 s; 60 °C 40 s; 40 cycles (only for the last two steps). The internal reference used in this experiment was *ACTIN2.* Data represent means of three replicates ± standard deviation (SD). The primers are listed in Appendix A.

For *Botrytis cinerea* stress, the spores of *Botrytis cinerea* were diluted with a certain amount of sterile water, and shaked it well to make the bacterial suspension concentration of *Botrytis cinerea* tomato be 10^7^ CFU /mL. The tomato leaves were sprayed with a suspension of 10^7^ CFU/mL of *Botrytis cinerea*. Care was taken to make sure that the front and back of the tomato leaves had bacterial suspension on them. The plants were then placed in a high humidity environment for the growth of spray water on tomato leaves keeping the other conditions unchanged. The sampling time points were 0, 0.5, 1, 3, 9, 12, 24, 48, 72 h. Under drought or ABA stress, the rhizosphere of potted plants were washed and put into 1/2 Hoagland culture medium (which contains calcium nitrate 945 mg/L, potassium nitrate 607 mg/L, ammonium phosphate 115 mg/L, magnesium sulfate 493 mg/L, iron salt solution 2.5 mL/L, trace elements 5 mL/L) to adapt to the growth for two days. Then the plants were transferred to 1/2 Hoagland medium containing 20% PEG6000 or 150 μM/mL ABA and the culturing was continued. The sampling time points were 0, 0.5, 3, 9, 12, 24 h.

### 5.4. Virus-Induced Gene Silencing (VIGS)

Agrobacterium *EHA105* were transformed with *pTRV2-SlTPR2* or *pTRV2-SlTPR4* along with *pTRV1*, *pTRV2-PDS*. Tomato plants were vacuum infiltrated with each group of plasmid mixtures for 4 min using vacuum centrifugal concentrator. The bacterial liquid could be seen into the tomato leaves. The infected tomatoes were cultivated in the dark at 22 °C overnight, and then cultivated in a greenhouse at 21 °C, a humidity of 30% and a photoperiod of 16 h/8 h. After the positive control plants turned white to the greatest extent, the leaves of the test group and the control group were taken to perform qRT-PCR detection of the silenced genes [49].

### 5.5. Yeast Two-Hybrid Assay

The *SlTPR4* cDNAs were amplified and cloned into *pGADT7*, and *atpA* cDNAs were amplified and cloned into *pGBKT7*. A yeast two-hybrid assay was performed following Matchmaker GAL4 two-hybrid system 3 kit instructions (Clontech, Mountain View, CA, USA). Yeast transformants were grown on SD-Trp-Leu-His medium. Primers used for the constructs were listed in Appendix A.

### 5.6. Pull-Down Assay

The *SlTPR4* cDNAs were amplified and cloned into *pQE-80L* with a His tag. The *atpA* cDNAs were amplified and cloned into *pGEX4T-1* with a GST tag (GE Healthcare, Uppsala, Sweden). The pull-down assay was performed as described previously [42]. Primers used for the constructs are listed in Appendix A.

## Figures and Tables

**Figure 1 ijms-22-00758-f001:**
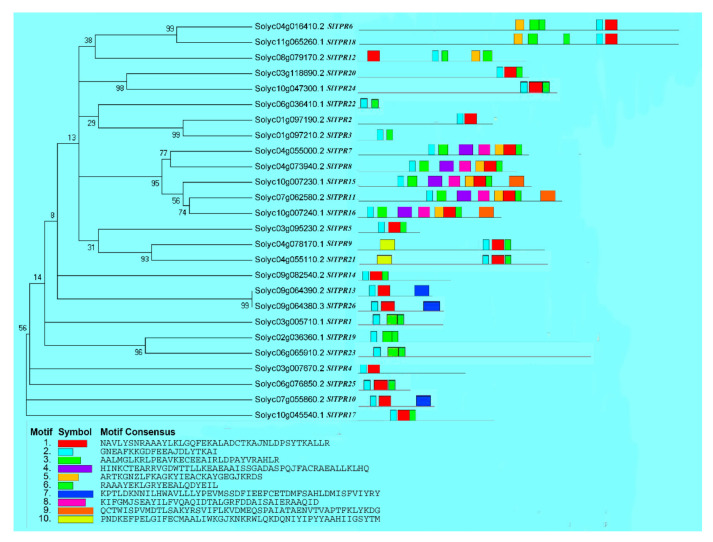
Conserved motifs in tomato *TPR* proteins. The protein sequences of 26 genes contain 10 conserved motifs, the same conserved motifs among proteins are represent with the same color.

**Figure 2 ijms-22-00758-f002:**
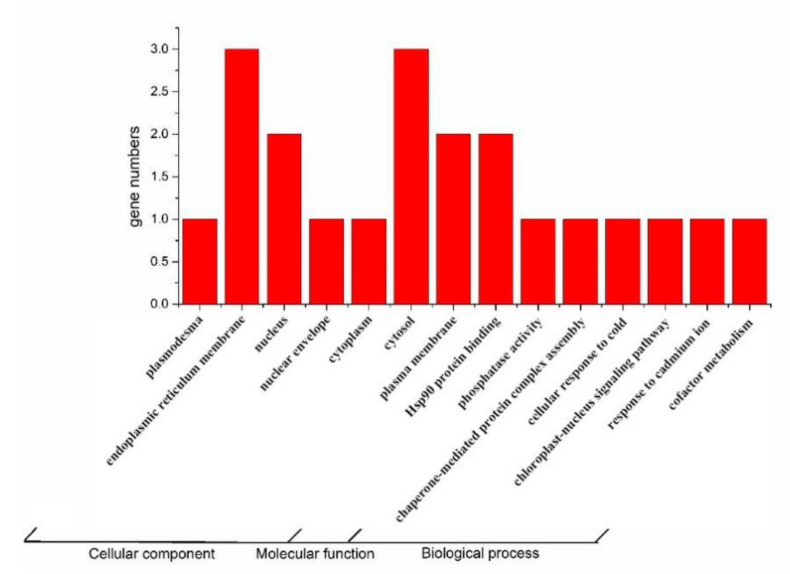
Gene ontology (GO) analysis output of tomato *TPR* genes. *SlTPRs* is involved in three kinds of biological processes, including 14 pathways including cellular component, molecular function and biological process.

**Figure 3 ijms-22-00758-f003:**
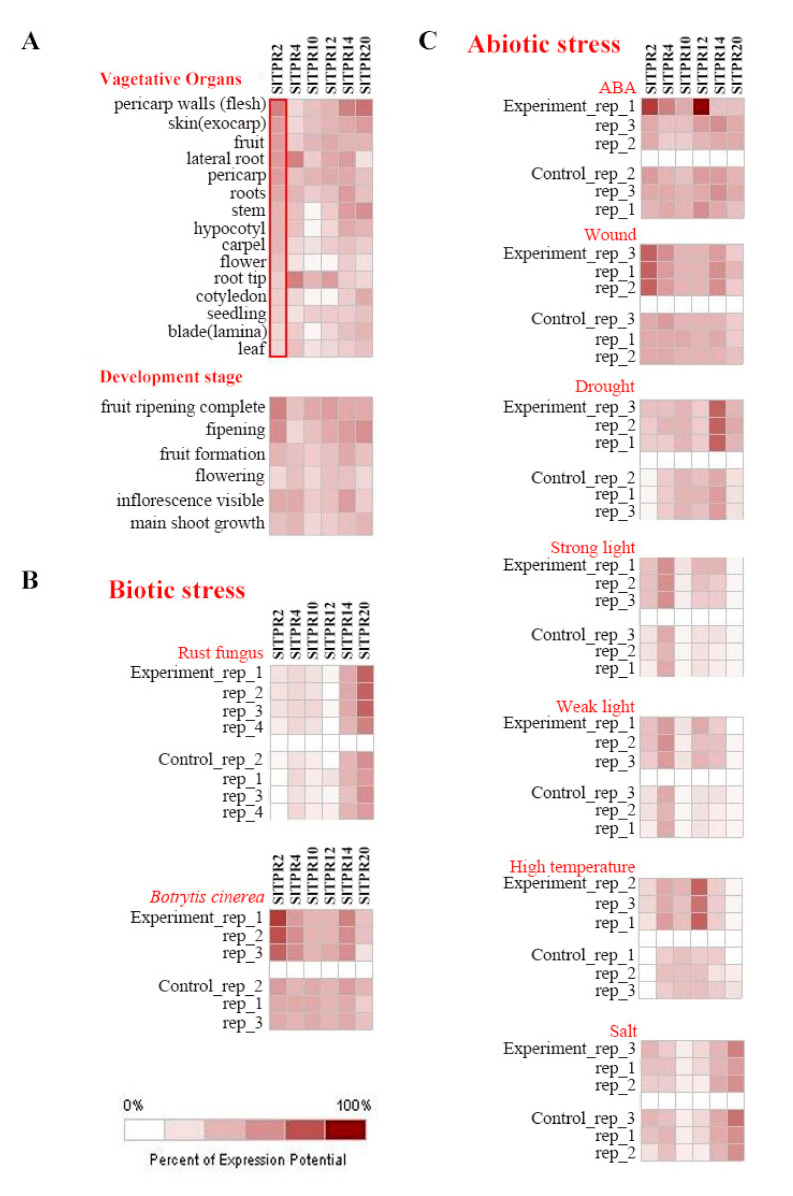
(**A**) Prediction and analysis of organizational positioning (**B**) Gene expression under biotic stress (**C**) Gene expression under abiotic stress. The expression pattern of the tomato *TPR* gene. We carried out 9 treatments on suitable tomato, including rust fungus, *Botrytis cinerea,* ABA, wound, drought, strong light, weak light, high temperature, salt. For rust fungus and *Botrytis cinerea*, the solution was diluted to 10^7^ cfu/ mL and sprayed on the 4-week-old seedlings. For ABA treatment, the concentration of the solution is 0.15 mM. For wound treatment, we use a punch to treat tomato leaves. All the four treatments were sampled at 72 h. For drought treatment, we sampled at 7 days. The strong light intensity was 1000 umol m^−2^ s^−1^, the weak light intensity was 200 umol m^−2^ s^−1^, and the control group was 450 umol m^−2^ s^−1^. Both treatments were sampled at 14 days. For high temperature treatment, samples were taken one hour after treatment at 40 °C. For NaCl treatment, the concentration of the solution is 100 mM and treatment were sampled at 24 h. All treatments were repeated at least 3 times.

**Figure 4 ijms-22-00758-f004:**
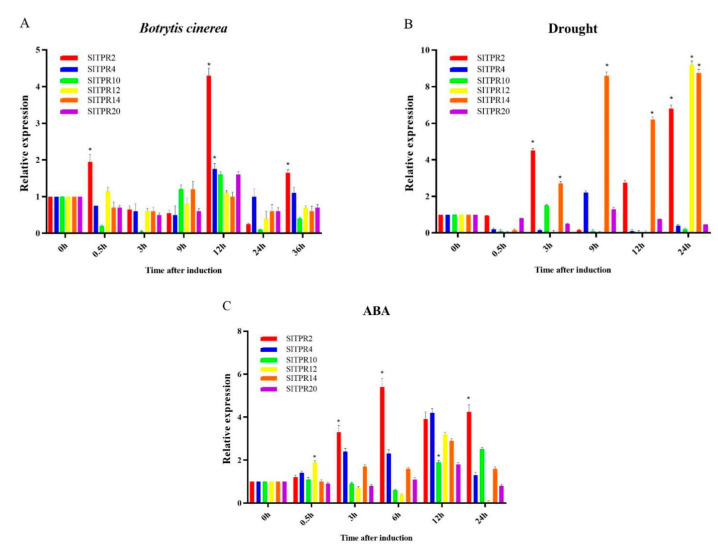
Expression analyses of tomato *TPR* genes under different stresses. Tomato seedlings were treated at 0 h with *Botrytis cinerea* (**A**), drought (**B**) and ABA (**C**) conditions and samples were collected for analysis at different times as indicated in the figure. The expression of each member of the tomato *TPR* gene family was detected by qRT-PCR. Columns are shown as the mean and the bars as the standard deviation of three biological and technical repeats. “*” represents a significant difference between the data point and control (*p* < 0.05).

**Figure 5 ijms-22-00758-f005:**
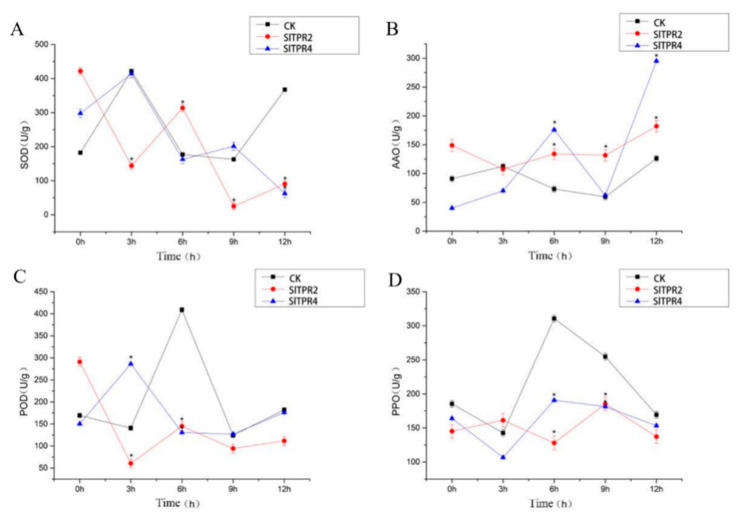
Effects of *B. cinerea* stress on the SOD (**A**), AAO (**B**), POD (**C**), PPO (**D**) of control and *SlTPR2*, *SlTPR4* silenced tomato. Error line indicate the standard deviation of three independent biological replicates. ANOVA was used for significance analysis. “*” represents a significant difference between the data point and control (*p* < 0.05).

**Figure 6 ijms-22-00758-f006:**
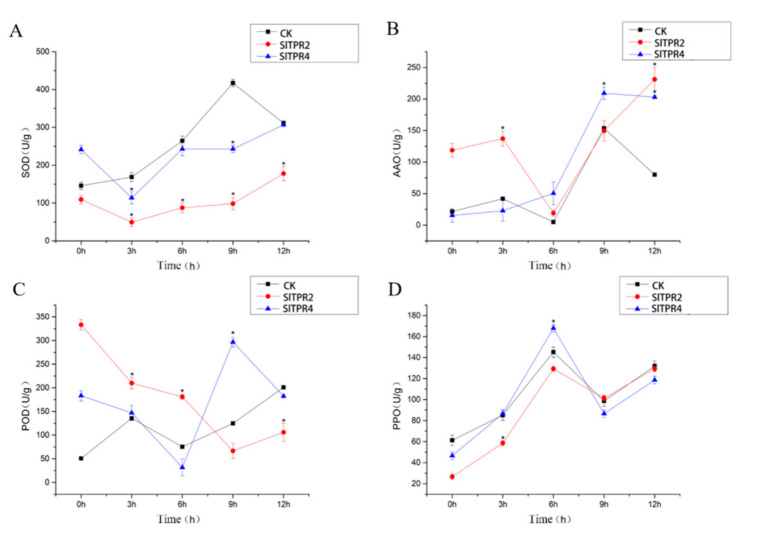
Effects of drought stress on the SOD (**A**), AAO (**B**), POD (**C**), PPO (**D**) in control and *SlTPR2*, *SlTPR4* silenced tomato. Error line indicate the standard deviation of three independent biological replicates. ANOVA was used for significance analysis. “*” represents a significant difference between the data point and control (*p* < 0.05).

**Figure 7 ijms-22-00758-f007:**
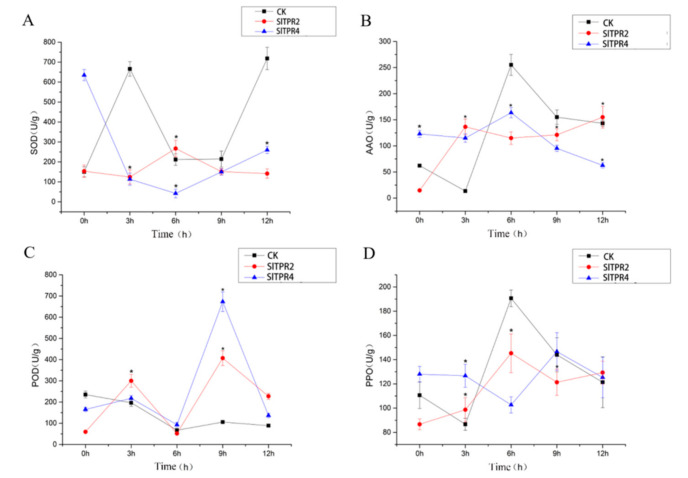
Effects of ABA stress on the SOD (**A**), AAO (**B**), POD (**C**), PPO (**D**) of control and *SlTPR2*, *SlTPR4* silenced tomato. Error line indicate the standard deviation of three independent biological replicates. ANOVA was used for significance analysis. “*” represents a significant difference between the data point and control (*p* < 0.05).

**Figure 8 ijms-22-00758-f008:**
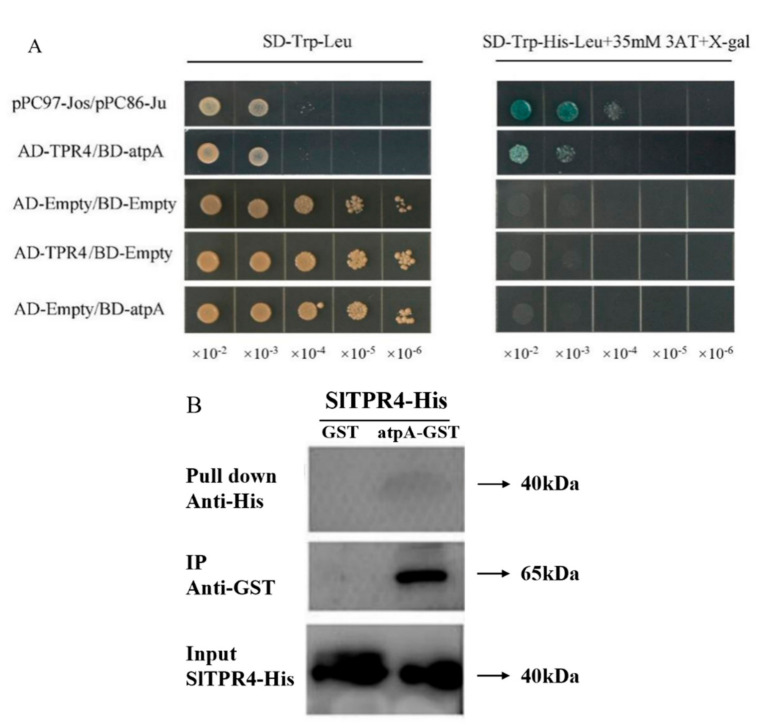
Identification of *SlTPR4* and *atpA* protein interaction. (**A**) Yeast cells co-transformed with different constructs were grown on selective media lacking Leu and Trp for 3 days to express TPR4 and atpA. Then the colonies were transferred to lacking Leu, Trp and His selection medium containing 35 mM 3-Aminotriazole (3AT) and cultured for three days to observe their growth. X-gal was added to the yeast grown in the three-deficiency medium and observe the blueness. pPC97-Jos/pPC86-Jun was used as a universal positive control. (**B**) Purified GST-atpA and GST proteins were incubated with TPR4-His, which were immobilized with Glutathione Sepharose 4B beads. The proteins were immunoprecipitated with anti-GST antibody and pulled down with anti-His antibody. The GST protein was included as a negative control. Three independent experiments were performed with similar results.

**Table 1 ijms-22-00758-t001:** Information of *TPR* gene family in tomato.

Gene Name	Sequence Accession	No. Amino Acid	Chromosome Location	pI	Protein M.W (kDa)	Subcellular Localization
*SlTPR1*	*solyc03g005710.1*	378	502006–505012	4.4	42,542	1
*SlTPR2*	*solyc01g097190.2*	464	88125140–8813834	5.11	49,913	1
*SlTPR3*	*solyc01g097210.2*	120	88147803–88150286	8.34	13,365	1
*SlTPR4*	*solyc03g007670.2*	370	2200919–2205536	5.11	41,245	1
*SlTPR5*	*solyc03g095230.2*	211	56209836–56215404	9.23	23,613	1
*SlTPR6*	*solyc04g016410.1*	1420	7200714–7218394	8.47	156,043	2
*SlTPR7*	*solyc04g055000.2*	587	53530227–53533608	9.26	64,012	3,4
*SlTPR8*	*solyc04g073940.2*	543	59949691..59951735	9.5	61,374	2
*SlTPR9*	*solyc04g078170.1*	644	62975642..62978282	8.92	73,681	1
*SlTPR10*	*solyc07g055860.2*	532	63773890..63783760	5.89	59,541	1
*SlTPR11*	*solyc07g062580.2*	701	65259956..65263961	9.05	76,013	2
*SlTPR12*	*solyc08g079170.2*	579	62804339..62810456	5.99	65,166	3
*SlTPR13*	*solyc09g064390.2*	256	61561701..61566057	6.34	28,958	1
*SlTPR14*	*solyc09g082540.2*	318	68259584..68265551	8.76	36,210	3
*SlTPR15*	*solyc10g007230.2*	596	1650247..1653367	8.72	65,786	2,1
*SlTPR16*	*solyc10g007240.2*	594	1660427..1662715	8.6	65,206	2,1
*SlTPR17*	*solyc10g045540.2*	470	34548304..34563332	8.83	52,399	2
*SlTPR18*	*solyc11g065260.1*	1261	50631990..50650877	6	138,554	4
*SlTPR19*	*solyc02g036360.1*	761	30576501..30578786	5.32	84,464	1,3
*SlTPR20*	*solyc03g118690.2*	590	67543799..67551645	8.94	64,110	3
*SlTPR21*	*solyc04g055110.2*	627	53710473..53724633	5.83	70,253	1
*SlTPR22*	*solyc06g036410.2*	661	25974062..25974262	4.95	7,364	1
*SlTPR23*	*solyc06g065910.2*	685	41307528..41309760	5.47	78,976	1
*SlTPR24*	*solyc10g047300.1*	598	40482669..40501656	9.2	65,597	2,3
*SlTPR25*	*solyc06g076850.2*	551	47742135..47745217	5.22	62,095	2
*SlTPR26*	*solyc09g064380.3*	128	61558744..61562593	6.34	28,958	1

Note: In subcellular localization: 1. cytoplasm; 2. outer membrane; 3. periplasm; 4. extracellular matrix.

## Data Availability

Data is contained within the article or Appendix A.

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
