# Peer review of "Identification and Functional Analysis of Tomato TPR Gene Family"

_ijms, 2021, doi:10.3390/ijms22020758_

Round 1

Reviewer 1 Report

The manuscript entitled " Identification and Functional Analysis of Tomato TPR Gene Family " sets out to investigate the TPR gene family in tomato under abiotic and biotics stresses. The authors reported that members of the tomato TPR family under study responded to various abiotic stresses and Botrytis cinerea stress, and the SlTPR2 and 25 SlTPR4 genes changed significantly under different stresses. They also reported that using VIGS technology to silence these two genes, the silenced plants showed reduced disease resistance. The study is on a topic of relevance and general interest to the readers of the journal. I found the paper to be overall well written and felt confident that the authors performed a careful and thorough experiment and spectral processing. However, I have several concerns about the presentation of the data that should be addressed before publication.

  • The authors are highly recommended to avoid using a personal pronoun (e.g., We, our, etc.); they can use the third party in the past tense's passive voice.
  • The authors need carefully to read through the manuscript to correct typos and grammars to improve the manuscript. In many places, they need to add spaces such as Line 159-160.
  • Any abbreviation must be associated with the full name at the first mention in the abstract and main text, then just use the abbreviation, such as GO in the abstract L. 22
  • In this manuscript a confusion about PCR terms; the author should use these terms as follow, Reverse Transcription Polymerase Chain Reaction (RT-PCR), Quantitative Real Time Polymerase Chain Reaction (qPCR), convention or regular Polymerase Chain Reaction (PCR).
  • Add a few sentences at the end of the introduction to summarize the aim of the study
  • Check the scientific names and make sure that they are in an italic format such as E. Coli in L. 281
  • The authors need to consider the term “resistance” vs “tolerance”, there is a big difference between both terms
  • 54-59: avoid using a short paragraph
  • 58-59: add a reference to this statement
  • 68 add a full name for SGT
  • 71-72: add a reference to this statement
  • 83: add a reference number to Kwon et al.
  • 109: do you mean 9.5 (SlTPR8)
  • 109: 13.365 and 156.043KDa, these numbers do not match Table 1, there is a misuse of the decimal, the number should be presented in the text and table 1 in this format 13,356 and 156,043
  • Table 1: round up the number in the Protein M.W. (kDa) and write in this format 42,452 for example
  • 128: remove and
  • Figure 1: add the gene next to the accession number
  • Figure 3 cation: there are so many typos; in the material and method section you, the author only mentioned Botrytis cinerea and here added rust fungus. Also, here they said NaCl concentration as 100 mM, but in the M+M section 200 mM, please check that.
  • 196: was 9 times, use was 9-fold instead and apply throughout the manuscript
  • Figure 4: move the panel label (A, B, C) to the upper left corner of the panel
  • Figure 4 caption: change “Values are shown as the mean ± standard deviation” to “ columns are shown as the mean and the bars as the standard deviation”
  • Figure 5, 6, 7, and 8: mark as A, B, C, and D and refer to that in the text when it is appropriate
  • 316: add a reference number to Srinivasa and add it to the list of reference as well
  • 318: add a reference number to Bhattarai
  • 329: add a reference number to Yoshiday et al.
  • 336: add a reference number to Koichi et al.
  • 340: add a reference number to Takken et al.
  • Add a conclusion section after the discussion section
  • 356: remove absolute
  • 357: what media did you use?
  • 364: add a reference for Hoagland culture medium
  • 391: add the reference or source for MEGA5.0 software
  • 400-401: add the source or reference for Goanna2ga program and Goslim Viewer
  • Section 4.3 need to be rewritten to differentiate between the PCR terms because it is confusing in the current format
  • 422: Agrobacterium should be in italic
  • Section 4.4 needs to be supported with reference

Author Response

Thank you very much for your efforts. We have revised the article. Please refer to the attachment for details.

Reviewer 2 Report

The paper describes the identification of 26 genes of the TPR gene family in tomato using bioinformatics based on tomato genome data. It also discusses their subcellular localization, phylogenetic evolution, presence of conserved motifs, expression profile in various tissues, GO analysis and their role in various biotic and abiotic stresses. The authors have generated useful data to support their observations. However, in many cases the data need to be communicated in an effective way (see my suggestions/comments below) for the readers to understand and follow.

In my opinion, the manuscript needs major revision for further reviewing of the work. I have provided here a list of comments that the authors could address while improving the manuscript. Hope you will find these useful.

  1. Line 18: ‘..cycle regulation, gene expression, degradation and other biological processes..’ degradation of what?
  2. Line 18-19: ‘The functions of TPR 18 gene in Arabidopsis and wheat plants has been well studied..’ Please replace ‘has’ with ‘have’.
  3. Line 22-24: ‘The qRT-PCR was used to detect the expression levels of each member of the tomato TPR gene family under biological stress (Botrytis cinerea) and abiotic stress (ABA, drought, high salt).’ what is ABA stress? It has not been mentioned in the main text as one of the stress conditions used in this study. Please explain.
  4. Line 62: ‘cycle regulation, gene expression, protein transport, and degradation…’ degradation of what?
  5. The introduction needs to be majorly revised and rewritten. In its current form, the introduction has lots of information, but it is not very well connected. It is hard to follow. Please try to focus it to the relevant information needed for the background of this work and revise accordingly so that it has a good flow.
  6. Please mention the full name of Sl. SlTPR has been mentioned many time, but the full name of Sl has never been mentioned in the main text.
  7. Table 1: Please explain what the numbers indicate in the column ‘subcellular localization’.
  8. Figure 2: The label of the last bar should be ‘cofactor metabolism’ instead of ‘cofactor metabolic’.
  9. Line 160: SlTPR14 and 20 have same gene numbers in the parenthesis. Please correct.
  10. Fig 1 and 3: Please mention TPR number instead of gene number as you did for Fig 4.
  11. Fig 3 has not been explained properly. The data for ‘Pathogen treatment’ and ‘Inducible treatments’ have not been explained in the main text.
  12. Line 192-194: ’Among the genes, SlTPR2 and SlTPR4 had the highest induction by Botrytis cinerea, ….control group, respectively.’ Fig 4A shows that SlTPR14 also had higher expression. Please clarify.
  13. Line 217: ‘..treated with Botrytis cinereal(A), drought(B) and high salt(C) conditions in different times.’ I think the treatment was done at 0 hr and samples were collected at different times for qRT-PCR. Please clarify.
  14. Line 224-225: ‘..it was found that SlTPR2 and SlTPR4 responded to various stresses more obviously.’ Please rephrase.
  15. What is CK in Fig S4?
  16. What are SOD, POD, PPO?
  17. Line 236-248: Fig 5, 6, 7 need better explanation. Please reanalyze the data and describe it properly.
  18. Line 274-275: ‘Previous report that ATP / ADP plays an important role in the initiation of ETI responses.’ Please cite appropriate reference. What is ETI?
  19. Fig 8A and B: i) what is line pPC97-Jos/pPC86-Ju? Please describe.
  20. ii) What are these conditions? SD-Trp-Leu and SD-Trp-Leu+35mM 3AT+X-gal? Please describe.

iii) What is SGT1?

  1. Line 278-283: ‘atpA was cloned…interact with atpA in vitro’ Please rewrite these results. Not very clear. What is pGADT7-SGT1/pGBKT17?
  2. The discussion section does not have good flow either. Also references are not appropriately cited. Please rewrite the entire discussion focusing on the results that you want to discuss. In its current form, there are lots of information which are not clearly connected with the appropriate result shown in this manuscript. It makes it harder to follow. Please try to stay focused on the discussion about what you actually want to communicate through your findings. Also the English in this section needs to be improved.
  3. At many places, the paper is hard to follow because of the English. Over all, the English for the entire manuscript needs to be improved a lot.
  4. In general, please mention the full form of any abbreviation when it has been introduced for the first time in the manuscript.

Author Response

(The authors gave the same response as above.)

Round 2

Reviewer 1 Report

The authors responded to all the comments and the manuscript is suitable for publication!

Author Response

Dear reviewer:
Thank you very much for your advice and contribution.

Reviewer 2 Report

The authors attempted making some corrections as suggested before. However, the manuscript still needs further improvement. The English of the overall manuscript still needs to be improved. In addition, some control data (please see my specific comments below) need to be included to support certain claims. Also, some data need to be represented in a better and more effective way (please see my specific comments below). The paper should be revised thoroughly and effectively for further consideration.

Comments:

  1. Introduction is still not satisfactory.i) Animal and microbial proteins having TPR motif has been discussed extensively. Since this study describes the identification of TPR gene family in tomato, so I would suggest to focus mainly on the plant proteins with TPR motifs and their role in biotic and abiotic stress response and keep the discussion about the animal and microbial proteins with TPR motifs brief and compact. ii) What is NB-LRR protein? iii) Relevant references are missing in any many cases during the discussion about the plant proteins in the induction. Please include the relevant references.
  2. Figure 2: Please make the edit in the original figure. At its current version, it doesn’t look like the edit has been done in the original figure and the ‘sm’ of the word metabolism has a different resolution than the rest of the word.
  1. Line 161: SlTPR14 gene number is still not correct.
  2. Figure 3: The pathogen treatment and inducible treatment sections of this figure has been compiled very poorly. It is labeled as pathogen 3 and pathogen, it does not indicate which panels are for rust fungus and which are for Botrytis cinerea . Secondly what do these labeling refer to: Pathogen 3_MoneyMaker_con_lea 11_rep_1 and Pathogen_Ailsa Craig_B. cin_mgreen_rep_1? I understand that rep 1,2, 3 correspond to different replica. But the rest of the labels are not understandable intuitively, in addition they are confusing. The inducible treatment panels don't have proper labeling as well. More over the second pair of panels under the Inducible treatments section say pathogen, which is incorrect as it cannot be an abiotic stressor. I would highly recommend to modify these sections properly so that these are more presentable and understandable for the readers. Currently the explanation of the data (line 171-180) for these sections cannot be followed since the data are not properly labeled.
  3. Line 198-199: you mentioned ‘Among the genes, SlTPR2, SlTPR4 and SlTPR14 had the highest induction by Botrytis cinerea, with an induction multiple of 4.5 and 2 times that of the control group, respectively.’ Please explain what you meant by induction multiple? Does it mean that the expression of a gene in the treated plant is x-times more than that in the control group of plants (ie. the non-treated plants)? However, none of the graphs in Fig 4 show the data for the control plants. Please include the data for the control plants in all the graphs in Fig 4. Without the data for the control plants, how do you know that that the expression pattern is due to the experimental treatment? Also, do you indicate the induction multiple is 4.5 for SlTPR2 and 2 for SlTPR4? Then what is the induction multiple for SlTPR14?
  4. Line 198-199: According to the data in Fig 41, the expression of SlTPR2 is actually decreased at 9h. Please clarify.
  5. Figure 4 legend: It is still not clear with the modification you made in the legend of Fig 4. Did you do the treatment at 0h and collect samples at different times for qRT-PCR analysis? If so, then modify the legend as ‘Tomato seedlings were treated at 0 h with Botrytis cinereal (A), drought (B) and high salt (C) conditions and samples were collected for analysis at different times as indicated in the figure.
  6. Line 227-228: It still is not very clear. In terms of the English, pPlease rephrase it as ‘After prediction by Genevestigator tomato gene chip platform and qRT-PCR results, it was found that under different stresses, the variation in gene expression is most pronounced for x, y, z genes.’ In terms of the data, the data for the control plants is critical for making any such claims. Therefore, I strongly recommend to include the data for the control plants in Fig 4.
  7. Figure S1 and S2: Use the gene name (SlTPR1, 2,..) in parenthesis along with the sequence accession number.
  8. Figure S4: Fig S4 should include the expression of SlTPR2 and SlTPR4 genes in the control group of plants. Currently, just including CK which is just wild type plants in the control group does not show the relative expression of these genes in this plant. Please include the appropriate data.
  9. Line 113-114: Please rephrase as ‘The analysis of tomato genome using bioinformatics has identified 26 candidate TPR genes (Table 1). It has been observed that the 26 SlTPR genes are...’
  10. Line 115-118: ‘Various physical and chemical properties of the corresponding proteins including their isoelectric point, molecular weight and amino acid number were analyzed using ExPaASy tool. The isoelectric point was found to range from 4.40 (SlTRP1) to 9.5 (SlTRP8). The molecular weight pf the proteins was found to range between 13,365 KDa (SlTPR3) and 156,043 KDa (SlTPR6).
  11. Line 132-134: Figure 1 also shows that the following genes also have the same conservative motif and order.

SlTPR22 & 3; SlTPR7 & 8; SlTPR15 & 11 & 16; SlTPR20 & 24 & 5 & 14 & 25 & 17; SlTPR9 & 21; SlTPR13 & 26 & 10; lTPR1 & 19 & 23; SlTPR2 & 4

  1. Line 147: ‘..and less in the intercellular filaments,..’ According to figure 2, it is plasmodesma instead of intercellular filaments.
  2. Line 151: ‘..and metabolism of coenzyme factor.’ Please replace coenzyme factor as cofactor.
  3. Section 2.4 Tissue Expression Analysis of Tomato TPRs Gene: What is Genevestigator tomato gene chip platform? Does figure 3 represent microarray data?
  4. Line 167: ‘..development of vegetative organ,’ Please rephrase as ‘..development of different vegetative organs.’
  5. Line 168: ‘blade (leaf lamina)’ Please replace as ‘blade (leaf lamina)’ as described in Fig 3.
  6. Line 232: ‘In order to further investigate the resistance of SlTPR2 and SlTPR4..’ Please replace ‘resistance’ with ‘role’.
  7. Line 242-244: ‘Under Botrytis cinerea stress, the content of SOD (Superoxide dismutase), POD (Peroxidase) and PPO (Polyphenol oxidase) in the silenced plants was significantly lower than that of the control group…’ The content of SOD did not decrease in SlTPR4 silenced plants compared to that of the control till 9 h post infection, rather it was comparable to the control. The content of SOD fluctuated in SlTPR2 silenced plants until 9 hours post infection and nothing can be concluded with respect to that of the control. However, the content of SOD decreased in both SlTPR2 and SlTPR4 silenced plants s at 12 h post infection compared to that of the control.
  8. Line 245-246: ‘The AAO (Ascorbic acid oxidase) content showed an overall trend of first

increasing and then decreasing.’ This statement does not match with the data presented in Fig 4B.

  1. Line 248-249: ‘..the silent group Gray mold 248 stress treatment (Figure 5).’ What is gray mold stress treatment?
  2. Line 252-253: ‘AAO can directly oxidize AsA (Ascorbic Acid Enzyme),..’ Please verify the substrate for AAO. AAO being an enzyme can certainly not oxidize ascorbic acid enzyme (AsA).
  3. Line 254-255: ‘The POD content of the control group was significantly higher than that of the silent group at various periods after the high salt treatment,’ It is not correlated with the data presented in Fig 7c.
  4. Line 259: ‘..SlTPR2 silenced line increases significantly at 6 h of the high-salt treatment,..’ According to Fig 7B, it is 3 h instead of 6 h.
  5. Line 280-281: ‘..the interaction protein of SlTPR4 was predicted using STRING and atpA which encodes the ATP synthase a subunit protein was identified.’ Please rephrase as ‘STRING was used to predict the interaction protein partener of SlTPR4. The protein, atpA, which encodes the ATP synthase α subunit was identified as the interaction partner of SlTPR4.’
  6. Line 285-291: Both the yeast two hybrid and the pull down experiments and their results need better description and explanation.

SD-Trp-Leu and SD-Trp-Leu+35mM 3AT+X-gal: what are these conditions? Please explain the rational for choosing these conditions and the expected experimental outcomes corresponding to these conditions.

In general, please describe the Y2H experiment in the context of your own constructs and the expected outcomes in your experiment. Then explain how the results corroborate your claims.

Similarly, pull down assay needs to be discussed in more detail.  It is hard to follow in its current state.

  1. The discussion has been edited, but it still needs further modification, especially improvement of the English. Please edit accordingly.

In addition some minor corrections are included below

Line 306: Please replace ‘location’ with ‘localization’.

Line 315: Please replace ‘Genestigator’ with ‘Genevestigator’.

Line 328: ‘BHATTARAI and other studies have shown..’ Only one reference (ref 390 has been given although it has been mentioned here that ‘Bhattarai and other studies have shown’. Also please replace ‘BHATTARAI’ with ‘Bhattarai’.

  1. It is nice that the authors have now added a conclusion section. However, it is hard to follow due to the English. Please consider rewriting it with professional English.
  2. Please revise the English of the ‘Materials and Methods’ section as well.

Author Response

Dear reviewers
Thank you very much for your advice and contribution. Please refer to the attachment for details.

Round 3

Reviewer 2 Report

The authors have made improvement on certain portions of the manuscript. However, it still needs further revision and clarification on some points as described below along with some other minor corrections. I appreciate that the authors annotated the revised portion in yellow for easy understanding during reviewing. I would also like to request the authors to kindly add a clean version of the entire manuscript along with the version that includes all the edits and corrections. It is otherwise not easy to read the entire manuscript with just the version containing all the edits and corrections.  

Please incorporate the corrections as accurately as possible. The paper should be revised very thoroughly and effectively for further consideration.

Comments:

  1. The introduction has been improved a lot.
  2. Line 79-80: ‘Deletion of SRFR1 results in plant 79 susceptibility’ Susceptible to what?
  3. Line 102-111: This paragraph talks about ATP synthase, but did not discuss its connection to TPR family proteins. It will be a bit abrupt to the readers. Please briefly mention the role of ATP synthase in the context of TPR family proteins.
  4. Line 29: ‘It was also showed’ Please replace ‘showed’ with ‘shown’.
  5. Line 134: ‘The molecular weight pf the proteins…’ Please replace ‘pf’ with ‘of’.
  6. Figure S1: The resolution of the text in this figure looks poor. Please try to improve the resolution of the texts used in this figure.
  7. Line 156-157: ‘SlTPR9 & SlTPR11 & SlTPR15’ It is not SlTPR 9, instead it is SlTPR 16 ie. SlTPR11 & SlTPR15 & SlTPR16.

Line 157: ‘SlTPR16 & SlTPR21’ Instead It is SlTPR9 & SlTPR21

Line 157: ‘SlTPR1 & SlTPR9 & SlTPR23’ Instead it will be SlTPR1 & SlTPR19 & SlTPR23

I did mention these group of genes in my earlier comments, still those have not been incorporated in the text correctly. Please do these revisions carefully.

  1. Figure S2: The resolution of the text describing the gene names and the rest of the figures is starkly different. Please make sure that the resolution of the entire figure along with the texts is uniform.
  2. Line 165-166: ‘The protein sequences of 26 genes contain 20 conserved motifs,..’ Only 10 motifs are shown in figure 1.
  3. Line 171-172: ‘and less in the plasmodesma, nucleus and cytoplasm’ Please replace ‘nucleus’ with ‘nuclear membrane’. Also, what is the difference between cytoplasm and cytosol in Figure 2?
  4. Line 195: ‘are highly expressed in the fruits.’ According to Figure 3, it is ‘pericarp walls’ instead of ‘fruits’.
  5. Line 203-204: ‘In the treatment of wound stress, the expression of SlTPR2 was significantly up-regulated.’ According to Figure 3, the expression of both SlTPR2 and SlTPR14 was significantly up-regulated in the treatment of wound stress.
  6. Line 203-204: ‘Under strong light and weak light, SlTPR12 and SlTPR14 increased significantly.’ According to Figure 3, the expression of both SlTPR4, SlTPR12, and SlTPR14 was significantly up-regulated under strong and weak light stress conditions.
  7. Figure 4: i) Panel A and B does not show the 0h graphs i.e. the graphs for the controls. ii) the color for each gene is not consistent between Panel A, B, and C. iii) Panel C: Please remove the gap between the bars representing the individual genes as done for Panel A and B. iv) the description line 226-233 is still not accurate according to the data presented in these panels. Please revise carefully and describe the data accurately as shown in this figure.
  8. Figure S4: Font of CK is bold while SlTPR2 and SlTPR4 are not in bold. Please be consistent for all the labels.
  9. Line 291-292: ‘….but from 9h to 12h, the significantly increase of AAO content in plant SlTPR4 silenced plants was more obvious.’ Please replace ‘the significantly increase of AAO content in plant SlTPR4 silenced plants was more obvious’ with ‘the increase of AAO content was more obvious in SlTPR4 silenced plants.’
  10. Line 324-325: ‘In addition, the change of POD content was irregular, but the content of PPO showed the same trend (Figure 6).’ Please replace this as ‘In addition, the change of POD content was irregular among the control and the two silenced lines, but the change of content of PPO showed the same trend between the control and the two silenced lines.’
  11. Figure 7: Is it ABA stress or salt stress? It is mentioned as ABA stress in the text (line 331-332) for the description of figure 7, but the figure legend says salt stress. Please clarify.
  12. Line 375: What is 3AT?
  13. Figure 8B: It is mentioned SGT1-His. It is mentioned later in the discussion that the homologous protein of SlTPR4 in Arabidopsis is SGT1. The description of the pull down assay (line 364-369) in the text mentioned that the interaction between SlTPR4 and atpA protein in vitro was tested by the pull down assay. Please clarify. If it is SGT1-His instead of SlTPR4, then please briefly describe in this section (line 364-369) and also in the figure legend of 8b (which are before the discussion section) what SGT1 is.
  14. Line 405-406: ‘Srinivasa found that the SGT1 protein in tomato and Arabidopsis can respond to the coronatine (COR) and jasmine secreted by P. syringae Acid (JA) and trigger plant immune response.’ Would it be jasmonic acid (JA) instead of jasmine? I think P. syringae is a pathogen, therefore, P. syringae Acid (JA) does not sound right. Please clarify and revise properly.
  15. Line 437: ‘..it was showed that SITPR4 interacts’ Please replace ‘showed’ with ‘shown’.
  16. Line 439: ‘Therefore, we suspect that TPR4 may enter..’ Please replace ‘suspect’ with ‘hypothesize’.
  17. Line 445-446: ‘It was showed that SlTPRs is an intron-rich gene family by structural analysis.’ Please replace ‘showed’ with ‘shown’ and ‘is an’ with ‘are’. The revised sentence would be ‘It was shown that SlTPRs are intron-rich gene family by structural analysis.’
  18. Line 446-447: ‘GO analysis predicted TPR genes has molecular chaperone, anti-cold, and Ca2+ response function.’ Please rephrase as ‘GO analysis predicted that TPR genes have molecular chaperone, anti-cold, and Ca2+ response functions.’
  19. The ‘Materials and Methods’ section is still not satisfactory. It should be written in passive voice though out this section. Currently it is not uniform. Please revise carefully.

Author Response

Dear reviewers
Thank you very much for your contribution to this article. We revised the article according to your suggestion. Please see the attachment for details.

Round 4

Reviewer 2 Report

My previous comments have been addressed properly. I have some minor comments as described below. The manuscripts can be accepted provided the authors address the following comments and incorporate the required modifications.

  1. Line 147-148 (Figure legend of Fig 1): ‘The protein sequences of 26 genes contain 20 conserved motifs, the same conserved motifs among proteins are represent with the same color.’ It should be 10 conserved motifs, not 20. Please rephrase the sentence as ‘The protein sequences of 26 genes contain 10 conserved motifs, the same conserved motifs among proteins are represented with the same color.’
  2. Line 295: ‘First, bind atpA-GST fusion protein or GST protein to Glutathione Sepharose 4B beads,’ Fig 8B says His beads (which should be Ni-NTA beads). Please clarify.
  3. Line 307-308: ‘..which were immobilized with His beads.’ It will be Ni-NTA beads if the His tagged protein was immobilized. However, the text (line 295) says GST-tagged protein was immobilized. Please clarify.
  4. Line 387-388: ‘When tomato to grow…experimental treatment’. Please rephrase this sentence as ‘The plant was used for experimental treatment when five leaves appeared on the plant (about a month).’
  5. Line-404-405: ‘…and determined up to 20 motifs.’ It is 10 motifs instead of 20 motifs according to Fig 1. Please rephrase as ‘..and 10 motifs were identified.’
  6. Line-423-426: ‘Added 10 ul 2X SYBR Premix ExTaq buffer, … and then ABI PRISM 7500 real-time quantitative PCR instrument was used for 425 PCR amplification.’ Please rephrase as ‘A mixture of 10 ul 2X SYBR Premix ExTaq buffer, 0.3 µL 10 mM 5 ' / 3'-Primer, 0.8 µL DyeII, 2 µL cDNA template, and double distilled water was added to a 96-well PCR plate to a final volume of 20 µL. Then ABI PRISM 7500 real-time quantitative PCR instrument was used for PCR amplification.’
  7. Line 427: ‘Datas represent means of..’ Please replace ‘Datas’ as ‘Data’.
  8. Line 429-431: ‘For Botrytis cinerea stress, added a certain amount of sterile water to the tomato overgrown with Botrytis cinerea to dilute it, and shaked it well to make the bacterial suspension concentration of Botrytis cinerea tomato be 107 CFU /mL.’ It is not clear if the tomato plants were grown with Botrytis cinerea or if the plants were grown normally without the pathogen and then the suspension of Botrytis cinerea was sprayed on the leaves. Please clarify.
  9. Line 431-435: ‘Treatment group: sprayed the Botrytis cinerea suspension with a concentration of 107 CFU / mL into tomato leaves, paid attention to ensure that the front and back of the tomato leaves had bacterial fluid, and placed it in a high humidity environment for the growth of spray water on tomato leaves, other conditions remain unchanged. The sampling time points were 0, 0.5, 1, 3, 9, 12, 24, 48,72 h.’ Please rephrase as ‘The tomato leaves were sprayed with a suspension of 107 CFU / mL of Botrytis cinerea. Care was taken to make sure that the front and back of the tomato leaves had bacterial suspension on them. The plants were then placed in a high humidity environment for the growth of spray water on tomato leaves keeping the other conditions unchanged. The sampling time points were 0, 0.5, 1, 3, 9, 12, 24, 48,72 h.’ Here please also include how you did the treatment for the control group.
  10. Line 438-441: ‘Treatment group: transfer the plants to 1/2 Hoagland medium containing 20% PEG6000 or 150 μM/mL ABA and continue culturing. Control group: transfer the plants to 1/2 Hoagland culture medium and continue culturing. The sampling time points were 0, 0.5, 3, 9, 12, 24 h.’ Please rephrase as ‘Treatment group: the plants were transferred to 1/2 Hoagland medium containing 20% PEG6000 or 150 μM/mL ABA and the culturing was continued. Control group: the plants were transferred to 1/2 Hoagland culture medium and the culturing was continued. The sampling time points were 0, 0.5, 3, 9, 12, 24 h.
  11. Line 443-445: ‘The constructed pTRV2-SlTPR2, pTRV2-SlTPR4 plasmids and pTRV1, pTRV2-PDS were transferred into Agrobacterium EHA105 together, using vacuum centrifugal concentrator to vacuum infiltrate and infect tomato materials, each group worked for 4 minutes.’ I have not understood the procedure very well from your description. Here is what I understood: ‘Agrobacterium EHA105 were transformed with pTRV2-SlTPR2 or pTRV2-SlTPR4 along with pTRV1, pTRV2-PDS. Tomato plants were vacuum infiltrated with each group of plasmid mixtures for 4 minutes using vacuum centrifugal concentrator.’ Is that what you actually did? If so, please rephrase your description with this. If not, please modify accordingly.

.

Author Response

Dear reviewer,
Thank you very much for your contribution to this article. Attached is our reply to the question, and the modified sentence has been marked in red in the original text.
